# Agents Explore the Environment Beyond Good Actions to Improve Their Model for Better Decisions

## Abstract

Improving the decision-making capabilities of agents is a key challenge on the road to artificial intelligence [25]. To improve the planning skills needed to make good decisions, MuZero's agent [17, 10, 1, 18, 13, 7] combines prediction by a network model and planning by a tree search using the predictions. MuZero's learning process can fail when predictions are poor but planning requires them [28]. We use this as an impetus to get the agent to explore parts of the decision tree in the environment that it otherwise would not explore. The agent achieves this, first by normal planning to come up with an improved policy [7]. Second, it randomly deviates from this policy at the beginning of each training episode. And third, it switches back to the improved policy at a random time step to experience the rewards from the environment associated with the improved policy, which is the basis for learning the correct value expectation. The simple board game Tic-Tac-Toe is used to illustrate how this approach can improve the agent's decision-making ability. The source code, written entirely in Java, is available at «a github url».

## 1 Introduction

A reinforcement learning agent has a simple interface to its environment [26, 25]: It partially observes the environment, acts, and receives rewards (Figure 1).

Despite this simplicity, it is hypothesised [23] that *intelligence, and its associated abilities, can be understood as subserving the maximisation of reward*.

Following this idea, MuZero [17] achieved a new state-of-the-art, outperforming all previous algorithms on the Atari suite and matching the superhuman performance of its predecessor AlphaZero at Go, Chess and Shogi. The MuZero agent learned acting through self-play, without even knowing the rules of the game - strictly following the agent-environment interface.

The agent's mind combines fast predictions from a neural network model with slow algorithmic planning. This is similar to the way humans use fast intuitive and slow rational thinking [11].

Despite MuZero's successes, its learning procedure can fail if the value prediction is poor where planning needs it. It has recently been shown how an amateur-level agent can beat KataGo [29, 30], a state-of-the-art open-source Go implementation based on MuZero's predecessor AlphaZero, by leading KataGo to parts of the decision tree that it would never visit in any self-play training games [28].

We use this as an impetus to make the agent curious about parts of the decision tree for which it otherwise gains little or no experience in the environment. We do not claim to provide a solution that solves all related problems, especially not the motivation example.

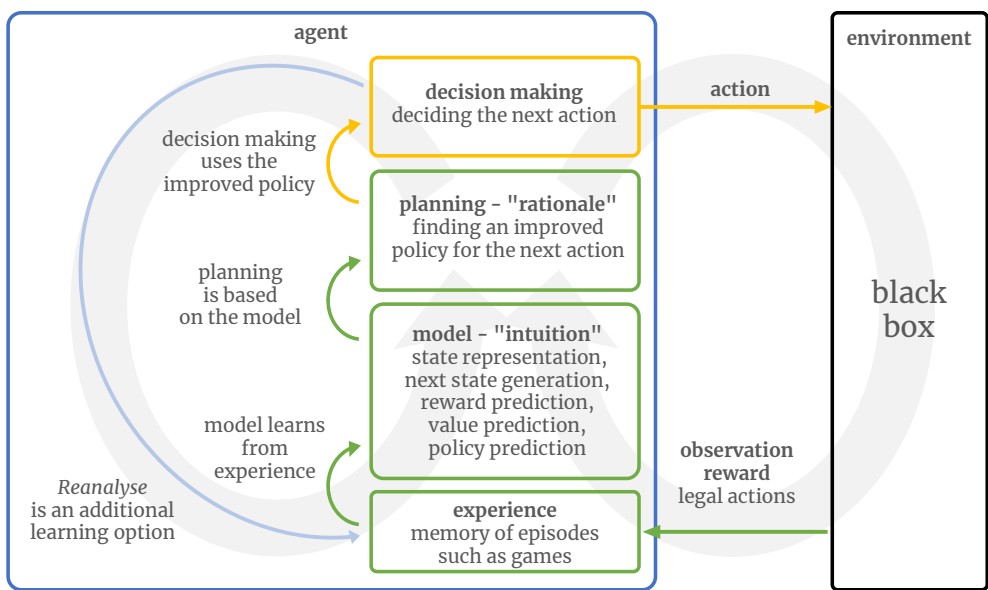

Figure 1: The interaction between the agent and the environment: The agent makes observations about the environment, is informed about the legal actions it can choose from, and potentially receives a reward after taking an action. This is the experienced information about the otherwise black-box environment. Together with internal information, such as actions taken, they form a memory of episodes. The agent uses this experience to train a model. The model's predictions include an in-mind state representation for observations, the value and policy for a state representation, the reward and the next state representation for an action. Based on the model's predictions, the agent plans an improved policy by partially unrolling the decision tree internally. Based on the improved policy resulting from the planning, the agent decides which action to take, taking into account its desire to explore the environment. The agent can also revisit states from its memory and re-analyse them [17, 18, 31]. With *Reanalyse*, there are two model optimisation loops: one via the environment and one entirely in the agent's mind.

Since the agent in this approach is actively seeking new experiences to feed into its model, we call it *curiosity*. We distinguish two domains of this *curiosity* - one for *known unknowns* and one for *unknown unknowns* [12, 4]. Our approach falls into the category of *curious about unknown unknowns*, as the agent seeks new experiences regardless of confidence in existing knowledge.

This active search consists of three parts: First, the agent performs normal planning at each time step, resulting in an improved policy. Second, in each training episode, the agent starts to act according to a policy that is steered by a temperature parameter $T > 1$ from the optimised policy received from planning ($T = 1$) towards a random action selection ($T \to \infty$). Third, to still learn the value of following the optimised policy from the associated environmental rewards, the agent randomly switches back to following the improved policy from planning. Thus, the action policy for all actions is a hybrid policy.

This makes the decision process a higher-level process that uses the tree search results from the planning process, but not necessarily on a one-to-one basis. So when we structure the agent, we add a *decision making* component that is responsible for deciding the next action, Figure 1. This responsibility includes, in particular, adding curiosity. With this structuring, we hope to contribute to the cross-disciplinary *quest for a common model of the intelligent decision maker* [25].

We also investigate two other cases with small contributions from us, arriving at three cases where we contribute - all three about the role of randomness:

**Additional randomness after planning** Use of the **hybrid policy** introduced here in the training context.

**Additional randomness before planning** In AlphaZero and MuZero, a **Dirichlet noise** was added to the prior probabilities in the root node when entering the tree search to aid exploration.

It was removed in Gumbel MuZero since it was not necessary to improve the policy with the model fixed. However, we use Dirichlet noise for the following heuristic reason: it adds a force toward choosing actions without unfair preference if the actions would not differ in value under a perfect strategy. If no force is added, one such action may be favoured by the agent, potentially preventing the agent from gaining experience from the parts of the decision tree after the unfavourable actions. This can lead to a worse model, a worse planning result, and therefore worse decisions. Another argument for avoiding unwarranted bias is to be stable against potential future changes in the environment that would favour an action other than the one the agent has learned to choose. We are aware that changing the policy with Dirichlet noise may cost some inference steps from the planning budget.

**Less randomness during planning in eager playout** Gumbel MuZero enters the planning for training playouts with the model's policy and draws from this policy - technically introducing a Gumbel value to achieve drawing without replacement. For the training context, this ensures that all root actions are considered exactly according to the existing knowledge of the agent. For an eager playout context, the situation is different. When making a decision only once, it can be beneficial for the agent to decide eagerly - like changing the temperature from 1 to 0. With this in mind, we also examine the playout case T=0 by setting the **Gumbel value to 0**.

We show for the simple board game Tic-Tac-Toe that these three contributions improve the decisions made by the agent. We use confidence intervals at the $99\%$ confidence level. In addition, we provide experimental examples to support our interpretation of how the improvements through using the *hybrid policy* and through using the Dirichlet noise occur - in these cases without proving statistical significance.

A limitation of this work is that we do not prove that we can reproduce all the results obtained by applying MuZero, in particular to the board games Go, Chess, Shogi and the Atari game suite.

Another limitation is that we do not show the application to the *Reanalyse* [17, 18, 31] loop here.

## 2 Recent Historical Background

AlphaGo [20] was the first AI engine to beat the best human player in a full-sized game of Go in March 2016. It used value networks to evaluate board positions and policy networks to select moves. The networks were trained using a combination of supervised learning from human expert games, and reinforcement learning from self-play games. The reinforcement learning used a tree search, which combines Monte Carlo simulation with value and policy networks.

AlphaGo Zero [21] eliminated the need to train with external input games. Thus, AlphaZero [22] generalised the AlphaGo algorithm and applied it to the games of Chess and Shogi. A major improvement to the algorithm was the continuous updating of the network.

In 2020, MuZero [17] has eliminated the need for a resettable simulator. Instead, MuZero learns a model of the environment to the extent necessary for its in-mind planning. It extends AlphaZero's successful application of the classic board games Go, Chess and Shogi to 57 Atari games. MuZero Unplugged [18] allows the agent to learn by re-analysing previously experienced episodes in mind.

Sampled Muzero [10] extends MuZero to domains with arbitrarily complex action spaces by planning over sampled actions. Stochastic MuZero [1] extends MuZero's deterministic model to a stochastic model that incorporates after states. It is demonstrated in the games 2048 and Backgammon.

EfficientZero [31], based on MuZero Unplugged [18] and SPR [19] achieved above-average human performance on Atari games with only two hours of real-time gaming experience. This experience efficiency was a milestone. Main contributions are (1) *Self-Supervised Consistency Loss*, (2) *End-To-End Prediction of the Value Prefix*, (3) *Model-Based Off-Policy Correction*. EfficientZero's source code is available on GitHub.

While MuZero's planning step produces an asymptotic policy improvement when many steps are used to unfold the decision tree, Gumbel MuZero [7, 6] introduced a planning algorithm that could improve the policy for any budget of unfolding steps - using a given model. The source code for the tree search is available on GitHub.

As a commercially relevant use case, MuZero has been applied to video stream compression [13]. And as the first extension of AlphaZero to mathematics, AlphaTensor [8] demonstrates the ability to accelerate the process of algorithmic discovery by finding faster matrix multiplication algorithms.

The open-source community has applied the AlphaZero and MuZero algorithms to various projects. Notable examples in the field of board games are Leela Chess Zero [14] and KataGo [29, 30].

The existence of open-source implementations encouraged the search for weaknesses in the agents. It was shown how adversarial policies could beat professional-level KataGo agents [28] using a strategy that an amateur player could follow. The main idea of the strategy is to lead the KataGo agent into areas of the decision tree where it has a poor value premise and therefore makes weak decisions.

## 3 Related Work

Finding an appropriate trade-off between exploration and exploitation is a core challenge in reinforcement learning [26]. By building a model that includes dynamics, as in MuZero [17], the magnitude of this challenge has increased because there are two worlds on stage: The environment as the real world and the model as an in-mind world. Gumbel MuZero [7] brought a planning algorithm that monotonically improves the policy with any budget of recurrent inference steps within MuZero's given in-mind world.

AlphaZero [22] and MuZero [17] add a Dirichlet noise to the model's policy predictions before starting the tree search in their planning step to ensure exploration.

The *off-policy maximum entropy deep reinforcement learning algorithm* SAC [9] uses the entropy of the policy times a temperature factor as an additional reward. Adding such an additional reward falls into the category of *curious about known unknowns* as this intrinsic reward is derived from the agent's policy.

After planning, MuZero [17] uses a temperature parameter $T$ to vary between $T = 1$ for exploration and $T = 0$ for exploitation following AlphaZero's [22] approach for board games. For Atari games, this is done for all moves, not just the first few moves as in board games. The temperature is lowered as a function of the number of training steps of the network, thereby shifting the planning policy from exploration to exploitation.

Go-Exploit [27] based on AlphaZero samples the starting state of its self-play trajectories from an archive of *states of interest*. This approach can only be used if the environment interaction allows episodes to start from any state.

## 4 What This Work Builds Upon

This work builds on MuZero [17]. For our examples, we use the case of non-intermediate rewards from the environment. For the planning component and the model base we use Gumbel MuZero [7]. The model is extended for *Self-Supervised Consistency Loss* from EfficientZero [31]. The resulting model is presented in Appendix A.

## 5 Illustrating the Need for Improvement of the Agent

The agent's need to explore the decision tree beyond good actions can be illustrated by the simple game of Tic-Tac-Toe [3].

In Tic-Tac-Toe, the optimal outcome for both players is a draw. Figure 2 shows such a game.

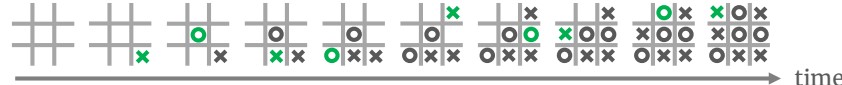

Figure 2: An example of an optimally played game of Tic-Tac-Toe.

Suppose an agent takes the role of both players - self-play - and only makes perfect moves in the environment. Then he would never observe from the environment what could happen after a bad move, e.g. after the first bad move shown in Figure 3.

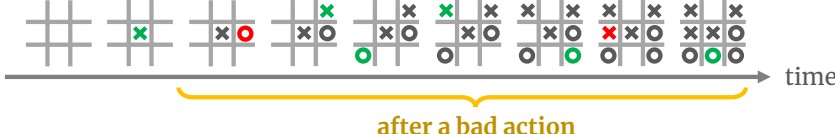

Figure 3: A Tic-Tac-Toe game with two bad actions (red), one by player o and one by player x.

Suppose such an agent takes the role of player x and plays against another player o. If a player o makes a bad move, the agent may not be able to take advantage of it and win. Instead, the agent might make a bad move as in Figure 3 and lose.

To observe more than the world of perfect actions, the agent must deviate from the perfect game when it acts in the environment during training. This could be achieved by separating the search for an optimised policy for the next action from the decision of what to do next. During training the decision component shown in Figure 1 could deviate from its optimised policy to get to novel parts of the decision tree and finish from there according to its optimised policy.

## 6 Agent Improvements

Two of the three contributions of the paper mentioned in the introduction are described here in more detail - supported by small proofs in the Appendix C.

### 6.1 Exploring Using a Hybrid Policy

Suppose we have a *normal policy* $P_{normal}$ and an *exploring policy* $P_{explore}$. Also, suppose the model is to be trained using $P_{normal}$. In a playout with $P_{explore} \neq P_{normal}$ there would be an off-policy-issue for the value target [18]. To avoid this problem, the playout could be done with a *hybrid policy* $P_{hybrid}$, starting with $P_{explore}$ and switching to $P_{normal}$ at a random time $t_{startNormal}$ before the expected end of the episode $t_{end}$.

$$P_{hybrid} = \begin{cases} P_{exploring} & \text{if } t < t_{startNormal} \\ P_{normal} & \text{if } t \geq t_{startNormal} \end{cases} \tag{1}$$

$$t_{startNormal} = random(0, t_{end}) \tag{2}$$

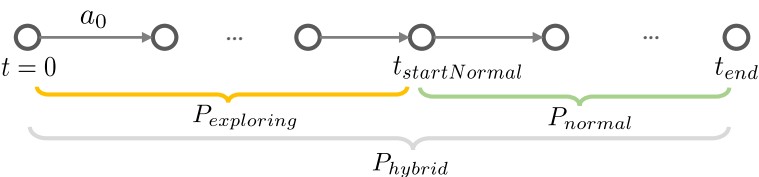

Figure 4: Before $t_{startNormal}$ the actions are chosen according to the *exploring policy* $P_{exploring}$. From $t_{startNormal}$ the actions are chosen according to the *normal policy* $P_{normal}$.

The value target after $t_{startNormal}$ could then be set up just as without exploration as the value information propagates backwards in time during training and is therefore not influenced by what happened before $t_{startNormal}$. But the value target before $t_{startNormal}$ would be set to keep its existing value. Therefore, the value function would only learn from the *normal policy*.

We concretise $P_{hybrid}$ in two steps. In the first step, we specify $P_{exploring}$ as a *drawing from a probability distribution*

$$\mathbf{p}_{exploring} = \text{softmax}\left(\ln(\mathbf{p}_{normal})/T\right) \tag{3}$$

with a temperature of $T > 1$.[1] $\mathbf{p}_{normal}$ is used for the models policy training target.

In the second step, we concretise $\mathbf{p}_{normal}$ to be the improved policy of Gumbel MuZero derived from the completed Q-values in the notation of Gumbel MuZero [7]. Using equation 21 in Appendix C.1 we get

$$\mathbf{p}_{exploring} = \text{softmax}\left(\frac{logits + \sigma(Q_{completed})}{T}\right) \tag{4}$$

The value target for the non-intermediate reward case is then given by

$$v_t^{target} = \begin{cases} v_{initialInference,t} & \text{if } t < t_{startNormal} \\ r_{t_{end}}^{measured} & \text{if } t \geq t_{startNormal} \end{cases} \tag{5}$$

where $r_{t_{end}}^{measured}$ is the reward returned by the environment and $v_{initialInference}$ is the value $v_t^0$ produced by the model version that is used when acting in the environment. This ensures that the value for this model version is not forced but later model versions taken from a buffer are forced towards $v_{initialInference}$. See Appendix D.5 for why we did not use the improved value from planning as the target value.

## 6.2 Eager Playout without Gumbel noise

When planning according to Gumbel MuZero [7] in an eager playout, we can enter planning with a temperature $0 \leq T \leq 1$ and still improve the decision by planning as shown in Appendix C.4. This is especially true for $T \to 0$ which we achieve by setting the Gumbel value to 0 (see Appendix C.3).

# 7 Experiments - Game Tic-Tac-Toe

The paper's three contributions are tested on the game Tic-Tac-Toe. Appendix D informs about experimental details, Appendix E about the open source implementation used to run the experiments.

## 7.1 Training With and Without Exploring - All Games

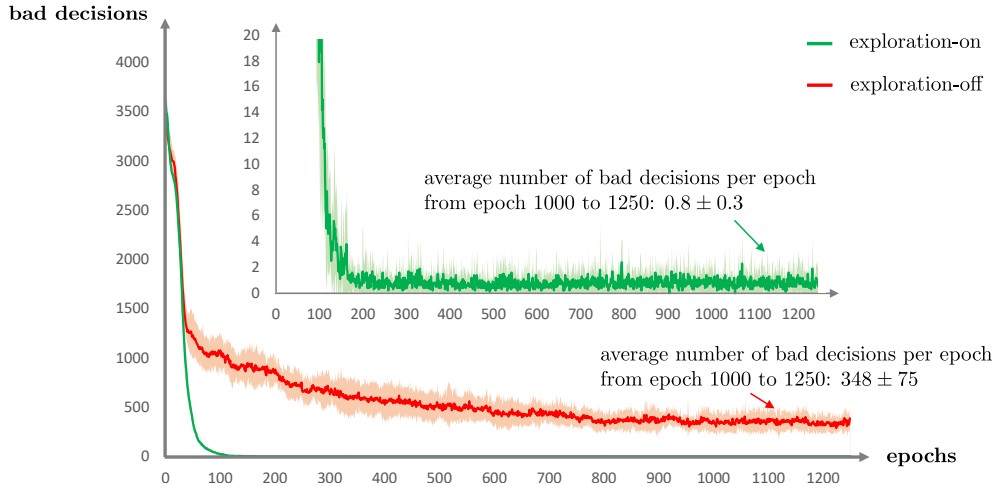

Figure 5: Number of bad decisions as a function of the training epoch - mean value over 10 samples with $99\%$ confidence intervals. For details on the counting of the bad decisions see Appendix D.

In Tic-Tac-Toe, the agent shows a large difference in the quality of decisions depending on whether the exploration introduced in the previous section is turned on or off. While without this exploration

---

[1]Note that the temperature parameter $T$ in MuZero [17] varies between $T = 1$ for exploration and $T = 0$ for exploitation, whereas here we explore with a temperature of $T > 1$.

the average of bad decisions for a trained model applied to all possible game decisions is $340 \pm 80$ after 1000 epochs, with exploration, it is $0,8 \pm 0,3$ (see Figure 5). This is an **improvement by a factor** $435 \pm 190$.

## 7.2 Training With and Without Exploring - One Game

To gain insight into the cause of the effect seen in Figure 5, we now restrict our investigation to the particular game shown in Figure 3. Note that the second move in this particular game is already a bad move, so all the states behind that move would not occur in perfect play.

From the model versions trained without exploration, we look for a model version with which the agent would make the second bad decision in the situation of Figure 3. To find out why it does so, and why the agent using a model trained with exploration would not, we look at the value expectation of the model $v_t^\tau$, since planning relies heavily on the quality of the value expectation. $t$ denotes the time at which the initial inference starts and the in-mind time $\tau$ denotes the number of recurrent inference steps from that point. For a detailed definition of $v_t^\tau$, see Appendix A.1.

When examining the value expectation of the model, it is not sufficient to consider the value expectation $v_t^0$ immediately after the initial inference. Since the unfolding of the decision tree happens at the in-mind time $\tau$, we need to look for all relevant in-mind times $\tau$.

Therefore we examine

$$v_t^\tau(t', t_{start}) = \begin{cases} v_{t'}^0 & \text{if } t' < t_{start} \\ v_{t_{start}}^{t'-t_{start}} & \text{if } t' \geq t_{start} \end{cases} \tag{6}$$

with $0 \leq t' \leq 18$ and $0 \leq t_{start} \leq 8$.

In Figure 6 we examine what the expectations of a particular model $v_t^\tau$ look like.

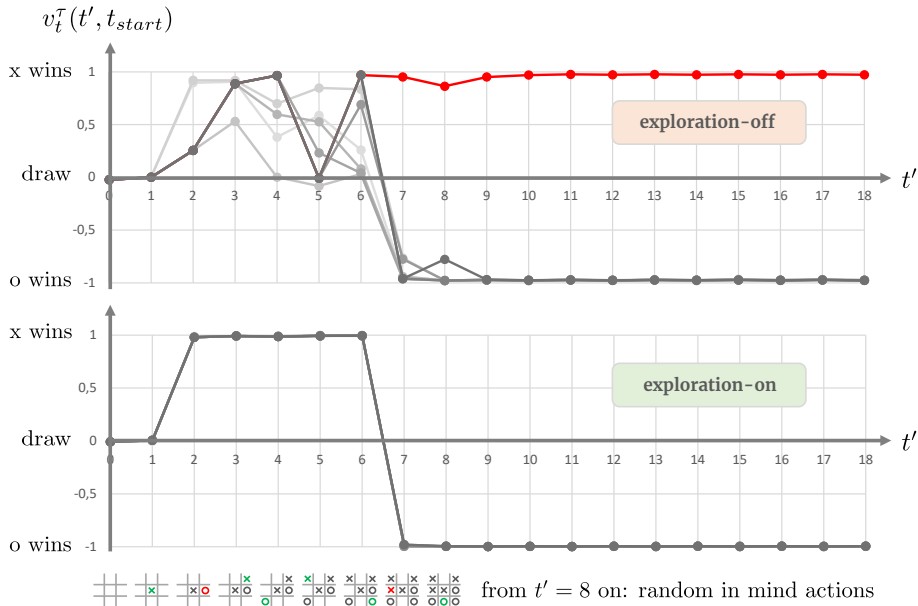

Figure 6: model version $epoch = 1028$, $t_{start} = 0$ in light grey to $t_{start} = 8$ in dark grey - for the exploration off case $t_{start} = 6$ in red. The red value expectation falsely pretends that player x's bad move would be good.

This is an example of a plausible cause - no statistical significance is claimed - that prevents agents trained without the additional exploration from making correct decisions: The value expectation of the model provides wrong values. The planning that uses them has no chance of leading to a correct decision.

## 7.3 Playout With and Without Gumbel Noise - All Games

The playouts during the test in Figure 5 were done with the same Gumbel value as during training. Playing out eagerly by setting the Gumbel value to $0$ during planning reduces the number of bad decisions. Figure 7 plots the difference *number of bad decisions with Gumbel minus the number of bad decisions without Gumbel*. In the exploration-on case, we see an **improvement by a factor** $2.1 \pm 0.3$.

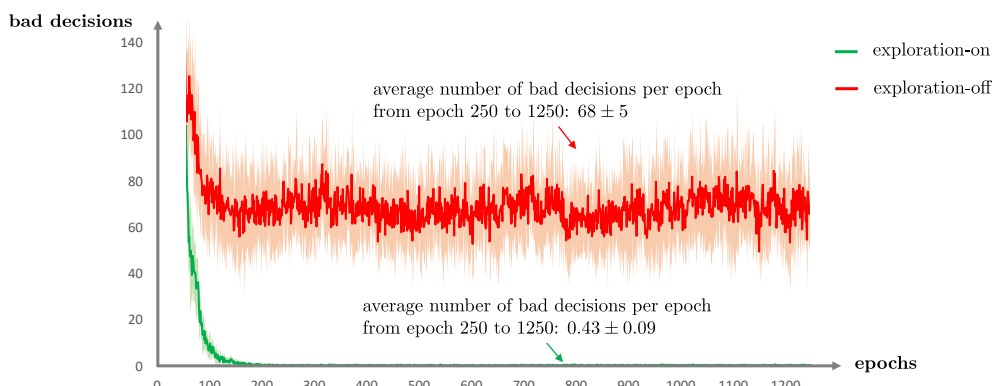

Figure 7: Number of bad decisions in playouts with Gumbel minus the number of bad decisions in playouts without Gumbel - a rolling average of the last 50 epochs, mean value over 10 samples, $99\%$ confidence intervals.

## 7.4 Training With and Without Dirichlet Noise - All Games

Figure 5 is based on models trained with Dirichlet noise added to the policy entering the tree search. We compare the decision performance of these models with models trained without Dirichlet noise, leaving all other hyperparameters unchanged. In the exploration-on case, we see an **improvement by a factor** $3.6 \pm 1.2$ using Dirichlet noise, see Figure 8 compared to Figure 5. Appendix D.7 speculates why this is the case.

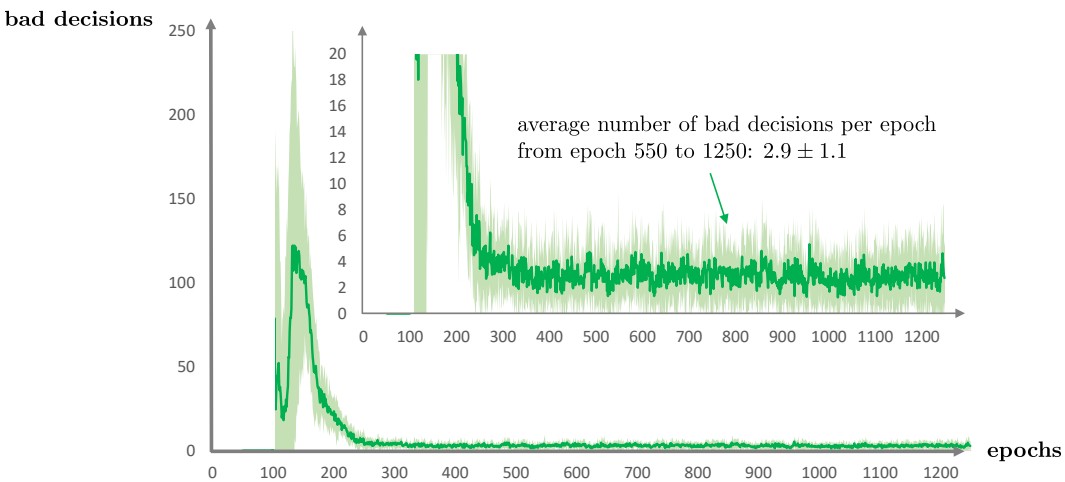

Figure 8: Number of bad decisions for models trained without Dirichlet noise minus the number of bad decisions for models trained with Dirichlet noise - rolling average of the last 100 epochs, 10 samples, $99\%$ confidence intervals.

## 8 Discussion

We have introduced a new exploration approach to learn more from the environment. The new idea is to use two separate policies in a combined hybrid policy $P_{hybrid}$, starting episodes with one for exploration $P_{exploring}$ - to take the agent to situations it would otherwise not experience - and randomly switching to the other policy $P_{normal}$ for finishing the episode with normal training. We derived $P_{exploring}$ from $P_{normal}$ by using a softmax temperature to introduce noise, set $P_{normal}$ to be the improved policy from Gumbel MuZero [7] and applied it to the game Tic-Tac-Toe. In these experiments, at a statistical confidence level of $99\%$, we observe a reduction in bad decisions by a factor of $435 \pm 190$. A selective check suggests that the reason for the wrong decisions before introducing the new exploration approach lies in an incorrect value expectation of the agent's model.

In further experiments on the Tic-Tac-Toe game at a statistical confidence level of $99\%$, we observed that training with Dirichlet noise resulted in a network with better decision ability than training without Dirichlet noise and that playout of the trained network without Gumbel noise showed better decision ability than playout with Gumbel noise.

Having found large improvement factors for Tic-Tac-Toe, we should ask ourselves: have we reached state-of-the-art? For all situations where we could fully unfold the decision tree in a classical manner, we should consider perfect decisions as state-of-the-art. Therefore we have not fully reached the state-of-the-art for Tic-Tac-Toe.

It would be interesting to see how the approaches tested here for Tic-Tac-Toe pay off for Go, Chess, Shogi, and the Atari games on which MuZero was tested.

What could be done to improve the approach presented here?

**Exploration Level** When using the hybrid policy in the experiments, we used a fixed exploration temperature. In general, a higher temperature will distribute the agent's starting points for normal policy actions more randomly, whereas a lower temperature would keep the starting points closer to the best action path the agent could take. A strategy needs to be found on how to best set the exploration level to improve decisions.

**Entropy reward** Entropy as an intrinsic reward[9] could be added as a curiosity mechanism for *known unknowns*. We speculate that this - as one aspect - would remove the need for Dirichlet noise. The measurements from Appendix D.7 seem to point in this direction.

***Reanalyse* learning cycle** The use of the *Reanalyse* learning cycle is a key feature in reducing the need for interaction with the environment. Extending the use of the techniques presented here to the *Reanalyse* learning cycle would therefore be useful. It would be of particular interest to *Reanalyse* the states that lead to rewarded actions, as the reward is a direct input from the environment and the source of the derived value. We speculate that this could improve the model's value predictions and thus the quality of decisions. A theoretically sound solution to the off-policy problem would be helpful in this regard.

**Adversarial Exploration** If the agent randomly deviates from the optimised strategy during exploration, it is unlikely to get into the situations the adversarial player put it into in Go [28]. Therefore, it may be necessary to devise an exploration strategy using an adversary as a counterpart in such games.

We hope to provide a useful technique for better learning the value function of the model as a basis for better planning-based decisions by the agent. It could serve as a starting point to help the agent become more curious.

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
