# OpenReview forum: "Agents Explore the Environment Beyond Good Actions to Improve Their Model for Better Decisions"
_NeurIPS.cc/2023/Conference — Submitted to NeurIPS 2023_

### Official Review · Reviewer_qakF · 2023-07-04

**Soundness:** 1 poor
**Presentation:** 1 poor
**Contribution:** 1 poor
**Rating:** 1
**Confidence:** 5

**Summary:**

Model-based planning agents such as MuZero leverages a learned model of the environment dynamics to learn a policy to follow in the actual environment. By planning using the learned model dynamics, the agent may generate a stronger policy than when limited to only experience from the real environment. The drawback to this approach lies in the agent being forced into decision states it has not seen before. The agent's model will have incorrect value predictions for these states and the planning stage will result in weaker or sub-optmal policies in these states. This paper proposes an exploration scheme that ensures the agent explores sub-optimal regions of the search space to build better value estimates of the states within.

**Strengths:**

**Originality:** The ideas introduced in this paper have not been published before as far as I can tell.

**Significance:** The algorithm proposed by the authors is nominally significant. It is a naive implementation of the explore-first-then-exploit type of algorithms.

**Clarity:** The writing itself of the paper was fair but could use considerable improvement.

**Quality:** It is hard to assess the strengths in the quality of this paper.

**Weaknesses:**

First off, I think it needs to be said that this is not a paper that can be improved upon with changes in the writing or presentation. The algorithm introduced is only a minor change to an already existing algorithm and it provides very little new insight.

That being said, I do want to encourage the authors to continue learning and working in the field. For the authors, I would recommend attending workshop or course on technical writing and building a stronger background in RL and planning.

For the submission specifically, I have a number of critiques. However, I will touch on only a few because as I've said this is not a paper that I think can be reworked and resubmitted:

1) The introduction of the paper goes well into page 3. A strong problem statement and motivation section can go along way in emphasizing the importance of the work but the paper needs to focus more on the actual contributions.

2) A similar note on the background and related works section. A lot of it deals with the history of the body of the work and not their relation to this particular paper. Furthermore, the background section does not provide the actual background needed for the paper. A good background section typically formalizes the problem state; clarifies any assumptions being made; defines notation and how the problem is being modelled; and provides a primer on the literature specifically needed to understand the rest of the paper. This submisison fails in those regards.

3) It is difficult to understand the algorithm introduced because nothing has been defined.

4) The experiments are limited a single small domain and does not give any indication of how the approach scales to larger ones.

5) The experiments do not provide comparisons to other baselines, that is, to other algorithms that take the explore-first approach which are, in my opinion, a direct competitor to proposed algorithm.

**Questions:**

1) In equation 4, should this be the definition of $P_{normal}$ or are you defining $P_{exploring}$ twice?

2) On line 40, you initially restrict $T> 1$ yet $T$ is set to values outside of this range. What is going on?

3) Re Lines 45 - 49: Do agents not already make decisions about next actions?

4) What is Gumbel MuZero's policy or how is it learned?

**Limitations:**

The authors provide a list of limitations of their work.

---

> ### Author Rebuttal · Authors · 2023-08-07
>
> Thank you for your review.
>
> 1. Equation (4) is derived from equation (3) after concretising the policy p_normal to be the improved policy of Gumbel MuZero. The improved policy of Gumbel MuZero is described in detail in the Gumbel MuZero paper (see answer to question 4).
>
> 2. In line 40 we refer to training and the exploration policy $P_{exploring}$ (see Figure 4). In other contextes, there can be more eager uses of the policy and therefore temperatures may be chosen from the range $(0,1]$:
>
>    * In the original MuZero paper during training.
>    * During an eager playout $T\rightarrow 0$ is an option.
>
> 3. Of course, all RI agents make decisions about actions. The idea here is to have a separation of concerns and an architectural layering. "Planning" in the sense of having a model (as used in the MuZero paper) and simply asking for the best next action and an improved policy (as a probability distribution over the action space) that produces a better potential reward is a well-defined concern and a task in itself - the Gumbel MuZero paper elaborates on this task. Using the result of this planning could be as trivial as simply executing the proposed action, or it could be more complicated, like using a policy that takes the agent to an interesting place - perhaps one where it sees a lack of knowledge about the environment - and another where it tries its best to learn from that situation how its actions are rewarded from there.
>
> 4. Gumbel MuZero is described in detail in the [cited paper](https://openreview.net/pdf?id=bERaNdoegnO), which also provides an [example implementation](https://github.com/deepmind/mctx). "Gumbel AlphaZero and Gumbel MuZero, without and with model learning, respectively, are state of the art on Go, Chess, and Atari, and significantly improve prior performance in planning with few simulations".
> In the Gumbel MuZero paper, equation 11 gives the improved strategy.

---

> > ### Comment · Reviewer_qakF · 2023-08-17
> > **Thank your for your rebuttal**
> >
> > I thank the authors for their rebuttal and answering the few questions I had. However, as they do not address the main issues that cause the paper to suffer, my opinion remains the same and I will not be changing my score.

---

### Official Review · Reviewer_tFVD · 2023-07-05

**Soundness:** 3 good
**Presentation:** 2 fair
**Contribution:** 2 fair
**Rating:** 4
**Confidence:** 5

**Summary:**

The paper presents a method to improve the exploration of the MuZero agent in games. The authors propose an hybrid policy that mixes an exploratory policy and the optimized policy. The exploratory policy is meant to reduce brittleness of optimal policies.
The new method is demonstrated on Tic-Tac-Toe.

**Strengths:**

The paper addresses maybe the most important issue in decision making. The authors' approach to encouraging exploration in the decision tree addresses an important problem that could have significant implications if successful. Adversarial examples in games and self-play is certainly an interesting domain to investigate this.
The motivation is good, and the authors also evaluate their approach on a small game. The idea is simple and makes sense. The appendix is also nice, and the source code is included. Prior work is also well researched.
I like that the authors analyzed an example gameplay in Section 6.2

**Weaknesses:**

Section 2 Recent Historical Background

- For relevant work on adversarial policies, I like that you included “Adversarial policies beat professional-level 330 go ais. arXiv preprint arXiv:2211.00241, 2022.”. Another paper that investigates this issue and I think should also be included is then “Timbers, Finbarr, et al. "Approximate exploitability: learning a best response." Proceedings of the International Joint Conference on Artificial Intelligence (IJCAI). 2022.”
- When it comes down to AlphaZero extensions, a good addition in this section would be "Player of games." arXiv preprint arXiv:2112.03178 (2021).”, as it extends AlphaZero to imperfect information games


Paper sometimes does not read that well and individual sections sometimes feel bit disconnected (i.e. little connection to other sections in terms of semantics and flow). E.g. the word curiosity is only really used at the beginning, and my understanding is that authors really just mean exploration (and indeed exploration is used in the rest of the paper). Also note that "curiosity" clashes with prior work on Intrinsic motivation (Oudeyer et al., 2007; Schmidhuber, 1991, ...).


The idea is very simple, adding exploration at the beginning, which AlphaZero already does using different methods. I am not sure if the main evaluation in Section 7, namely Figure 5 is entirely fair (I might be wrong). The figure compares exploration-on vs exploration-off, but I don’t think that exploration-off in this case collapses to the “baseline” case? (e.g. MuZero or AlphaZero, as both do also force exploration at the beginning)


This issue connects to the second one, which is that the experiments are not that well explained. It is fine that the authors moved many details to the appendix (which contains a lot of interesting information), but the main body of the paper still should include enough for people to understand what is presented.


But my biggest issue is that the main challenge in balancing exploration in learning in games is when the approximator can’t really memorize the full game. If it could, than any exploration just works as we get to see all the states and memorize them (i.e. no generalization is needed). The game presented now is too small to see the effect, I suggest the authors run it on e.g. connect four. I believe the current experiments are simply not sufficient.


Finally, I think the measure of whether the exploration helps or not should ultimately be exploitability rather than just a uniform measure of bad decision over all the game states.



**Questions:**

Can you elaborate why you chose MuZero rather than AlphaZero to investigate the exploration issue? What’s the point of learning the model in your case?


Can you also measure exploitability for exploration on/off? I believe that is the ultimate metric in your case, and I also believe the suggested exploration would indeed help there.


How would the proposed method perform in more complex environments where the representational power of the estimator is limited with respect to the complexity of the full environment?


**Limitations:**

Addressed

---

> ### Author Rebuttal · Authors · 2023-08-07
>
> Thank you for your review. Your feedback is very helpful for us to improve.
>
> Our main task is to create a MuZero implementation – that is set for us. We are aware that in moving from AlphaZero to MuZero, the model needs to additionally learn to move forward in time, allowing the agent to operate in environments without a perfect simulator. If we had to choose the simplest solution to a given problem, given a perfect simulator, we would choose AlphaZero over MuZero. We are also aware that AlphaZero and MuZero play to their strengths in environments with a huge number of possible states where "the approximator can't really memorise the full game". In search of a simple and fast low-level integration test, we - somewhat naively - choose the game Tic-Tac-Toe. Our expectation as a test result for the game Tic-Tac-Toe is that there is not a single wrong decision in the whole decision tree.  We believe that by choosing Tic-Tac-Toe we have chosen a corner case where MuZero has limitations that we believe should be overcome in such a way that our Tic-Tac-Toe integration test passes, but certainly all MuZero's previous achievements in terms of effectiveness and - if possible - efficiency should also be reached. Although we have not achieved this goal here, we believe that we have taken a first step in this direction.
>
> Thank you for introducing us to the two papers on exploitability. We started to dive into them, added them to the history section of the paper, implemented a perfect exploiter for our trivial tic-tac-toe test, ran the generation of a Figure 5 type evaluation, and look forward to using a "best response approximator" for cases like Go5x5 (our results there look promising)  or Connect-Four. Aiming for exploitability as low as possible in multiplayer games is now a must for us. For the mentioned exploitations of go agents, it looks like the perfect metric.
>
> In our exploitability metric, which we have just implemented, we always start from the beginning of the game. We realised - maybe we need to gain a deeper understanding of the two papers - that this has the limitation that the exploited agent could hide a weakness by favouring a subset of otherwise equal best choices (e.g. instead of having equal probability on all possible first moves in Tic-Tac-Toe, the exploited agent X could always choose a particular first move and thereby hiding a weakness following one of the never chosen first moves.). This limitation could be seen as a weakness if one also aims for stability against small environmental changes - nothing that we have to face in board games. Zero exploitability is therefore a strong goal but not the best that we could aim for. For the tiny game of Tic-Tac-Toe we would like to aim for the best. One remark: Adding Dirichlet noise in our implementation at planning entry adds a force to equalise the probability between otherwise equal options - this at least helps to avoid running into the aforementioned weakness of the exploitability metric.
>
> Concerning your last question we could give some heuristics: Let us start with the configuration $T=1$ in equation (3). In this case, we are using the usual improved policy from planning to act in the environment and to train the model, but the value targets (equation 5) are only perfect for $t\geq t_{startNormal}$ and otherwise such that the value loss is essentially zero. We speculate that this makes the convergence simply slower, as policy learning stays the same and the loss force on the value expectation is partially the same as usual and otherwise zero.  When increasing $T$ in equation (3) the character of the policy training and the suboptimal value training stays the same, but we get additional exploring of the environment in the vicinity of the policy. So the possible additional forces should only come from the new experiences made in the additional exploration of the environment.

---

### Official Review · Reviewer_8jEf · 2023-07-06

**Soundness:** 1 poor
**Presentation:** 2 fair
**Contribution:** 1 poor
**Rating:** 3
**Confidence:** 4

**Summary:**

Inspired by the failure of KataGo against an amateur-level agent, the author proposes to use an additional randomization scheme to encourage the agent to explore the less experienced part of the decision tree. Such a randomization scheme allows the agent to randomly deviate from the planned policy, and then switch back to learn the correct value function. Empirically, they evaluate their method on Tic-Tac-Toe and justify the effectiveness of their method.


**Strengths:**

1. The problem of exploring the less experienced part of the decision tree is interesting.
2. The paper provides detailed related work.


**Weaknesses:**

The main concern of this paper is the contribution is marginal.

The proposed method is a minor modification (a randomization scheme) to the existing method. This is not to say the investigated problem itself is not important. But with such a minor modification, the paper needs to provide sufficient evidence to prove its effectiveness, either by theoretical analysis or a large body of experiments. However, I am afraid the paper provides neither. For the evaluation, the paper only evaluates the proposed method in a very simple scenario, Tic-Tac-Toe, which is way too easy. More experiments on more complicated scenarios are definitely needed.

The writing is unprofessional and should be improved substantially. For example, references should not be included in the abstract.



**Questions:**

n/a

**Limitations:**

Yes

---

> ### Author Rebuttal · Authors · 2023-08-07
>
> Thank you for your review.

---

### Official Review · Reviewer_8rVX · 2023-07-07

**Soundness:** 2 fair
**Presentation:** 2 fair
**Contribution:** 2 fair
**Rating:** 3
**Confidence:** 3

**Summary:**

To improve prediction accuracy, this paper divides the training phase's policy into exploration and normal policies. The proposed algorithm is tested in the game of Tic-Tac-Toe, and different noise strategies are introduced for experimentation.


**Strengths:**

The paper introduces a novel method for increasing policy exploration and experiments with different noise strategies for exploration.


**Weaknesses:**

1. The experiments lack persuasiveness as there is no comparison or theoretical analysis with advanced exploration algorithms.

2. The discrepancy between the mentioned "poor" predictions and the proposed exploration method is not fully addressed.

3. The setting of random time $t_{startNormal}$ is not clear.


**Questions:**

See the weakness part.

**Limitations:**

The application of the proposed algorithm in more complex tasks and its effectiveness compared to algorithms with other exploration techniques remains to be studied.

---

> ### Author Rebuttal · Authors · 2023-08-07
>
> Thank you for your review.
>
> 1. We use a proven very strong algorithm with well-founded and tested exploration strategies, namely Gumbel MuZero as the baseline. During the implementation, we were looking for an integration test - and by choosing Tic-Tac-Toe as such a test, we believe that we have taken the very strong algorithm Gumbel MuZero into a corner case where it could not play out its strength and instead has a weakness. The MuZero paper showed that the algorithm mastered three board games with a large number of states and that it mastered classic Atari games as an example of cases where the environment doesn't come with a perfect simulator.
> We believe that the strong algorithm of Gumbel MuZero should be adapted in such a way that it retains all its achievements, but does not produce a single wrong decision on the full decision tree of the trivial game Tic-Tac-Toe. We believe that we have made a step in this direction with this paper, but we have clearly not reached this goal yet.  The algorithm we use inherits from Gumbel MuZero in that the policy targets are set exactly the same way and the value targets are set the same way for $t\geq t_{startNormal}$ and otherwise such that the value loss is zero.  It is a bit like starting the usual Gumbel MuZero environment playouts from different points on the decision tree, but getting there from the start of the game and on the way to the start points still benefiting from the policy improvement by planning, but doing nothing about the value because the actions there are off-policy.
>
> 2. We are not sure if we have understood your second question correctly.
> With Gumbel MuZero we have a search algorithm that - with mathematical proof in the Gumbel MuZero paper - guarantees a potential policy improvement for any number of in-mind planning steps in the context of a given model. The weakness, however, is that if the model is not able to represent the reward situation in the real environment sufficiently accurately, then even an improvement in policy relative to the imperfect model may result in a worse policy relative to the real environment. An illustration of a particular case is given in Figure 6.
>
> 3. Before playing a game, the time $t_{startNormal}$ is chosen as a random number from the interval $[0, t_{end})$, where $t_{end}$ is taken as the maxGameLength from the games in the game buffer.

---

### Decision · Program_Chairs · 2023-09-21

**Decision:**

Reject

**Comment:**

Positives
+ Interesting, well motivated problem
+ good coverage of the literature
+ nice illustrative results on tic-tac-toe

Issues
1. the algorithmic change is very small, which is good, but puts increased emphasis on the evidence of the contribution
2. the evidence of the contribution is small: small scale experiments in one problem, no theory
--> in particular, there is an open question here: do the benefits of the new idea extend to the case where the agent cannot easily represent the task (where memorization is not possible i.e., at scale).
3. the paper does not measure exploitability, which is a key measure of whether the exploration helps or not
4. writing needs improvement & experiments are not well explained in the main text

The reviewers (including a senior reviewer with very relevant background) all agreed on reject and stated that the author response did not address or explain-away the key limitations listed above (1, 2, & 3).

The reviewers encouraged the authors to take the next steps and if successful the paper will be very strong! Some reviews gave nice suggestions and the authors seem interested in integrating them.